# Application of a Fusion Method for Gas and Particle Air Pollutants between Observational Data and Chemical Transport Model Simulations Over the Contiguous United States for 2005–2014

**DOI:** 10.3390/ijerph16183314

**Published:** 2019-09-09

**Authors:** Niru Senthilkumar, Mark Gilfether, Francesca Metcalf, Armistead G. Russell, James A. Mulholland, Howard H. Chang

**Affiliations:** 1School of Civil and Environmental Engineering, Georgia Institute of Technology, Atlanta, GA 30332, USA; 2Department of Biostatistics and Bioinformatics, Rollins School of Public Health, Emory University, Atlanta, GA 30322, USA

**Keywords:** spatiotemporal pollutant fields, data fusion, air pollution, *CMAQ*, particulate species, gas species

## Abstract

Accurate spatiotemporal air quality data are critical for use in assessment of regulatory effectiveness and for exposure assessment in health studies. A number of data fusion methods have been developed to combine observational data and chemical transport model (CTM) results. Our approach focuses on preserving the temporal variation provided by observational data while deriving the spatial variation from the community multiscale air quality (*CMAQ*) simulations, a type of CTM. Here we show the results of fusing regulatory monitoring observational data with 12 km resolution CTM simulation results for 12 pollutants (CO, NOx, NO_2_, SO_2,_ O_3_, PM_2.5_, PM_10_, NO_3_^−^, NH_4_^+^, EC, OC, SO_4_^2−^) over the contiguous United States on a daily basis for a period of ten years (2005–2014). An annual mean regression between the CTM simulations and observational data is used to estimate the average spatial fields, and spatial interpolation of observations normalized by predicted annual average is used to provide the daily variation. Results match the temporal variation well (*R*^2^ values ranging from 0.84–0.98 across pollutants) and the spatial variation less well (*R*^2^ values 0.42–0.94). Ten-fold cross validation shows normalized root mean square error values of 60% or less and spatiotemporal *R*^2^ values of 0.4 or more for all pollutants except SO_2_.

## 1. Introduction

Fine scale air pollution fields with complete spatiotemporal coverage are of importance to public health researchers who aim to understand the drivers of adverse health effects [1,2]. Health exposure studies involve matching ground monitor air pollution measurements to individuals within the nearby vicinity of an air pollution monitor [3,4,5]. These measured concentrations are then used to develop a city-wide health effect estimate [6]. However, this method is restricted by the limited spatial monitoring networks across the contiguous United States, the impacts of local emissions on a single monitor, and the temporal completeness of the air monitor data [7,8,9].

One method to address the sparseness and temporal incompleteness of air monitoring data uses chemical transport model (CTM) results. CTMs use the governing chemical and physical principals with meteorology and emissions data to simulate pollutant concentrations [10,11,12]. One widely used CTM is the community multiscale air quality (*CMAQ*) model. Although CTMs can create spatially and temporally complete pollutant fields, there are biases in the model results [13,14,15]. Such biases differ based on the species simulated and the domain [10]. Examples of this bias include over-predictions of particulate nitrate and ammonium and underpredictions of carbonaceous aerosols in the spring and summer [11]. For the ozone these biases include an underestimation in North America of daytime ozone in the spring and an overestimation of daytime concentrations during the summer [10,16]. Evaluation of *CMAQ* to the ozone monitoring instrument (OMI) found that *CMAQ* overestimates summertime nitrogen dioxide (NO_2_) in urban locations and underestimates in rural locations [17]. As *CMAQ* and CTMs improve and the biases are better understood, there is less of a need to modify CTM results for use in exposure studies [18,19].

CTM simulations can be combined with air quality observations through data fusion (DF) methods to other air quality data to correct for the biases associated with the model outputs [20,21]. Models can be fused through mathematical methods such as statistical interpolation or machine learning approaches [22,23,24,25,26,27,28]. The data fused with CTM outputs can range from satellite, observational, land use, or meteorology data. Interpolation methods such as ordinary kriging and inverse distance weighting (IDW) can interpolate a parameter over space, while machine learning methods such as neural network, random forest, and decision trees create a model using a set of preselected variables [29]. The method selected depends on the pollutant being studied and the type of data available [30,31]. For example, speciated particulate matter may only have observational data available, while particulate matter less than equal to 2.5 microns (PM_2.5_) has satellite data derived aerosol optical depth (AOD) and observational data available [32,33,34]. Each method is associated varying performance depending on the type of data and method being used [35].

The approach of this study is to use a statistical DF method to combine observational data to the *CMAQ* model simulations to create accurate spatiotemporally resolved air pollution fields. We modified an existing data fusion method (Friberg et al) and applied it regionally across the contiguous United States at a 12 km resolution for 12 pollutants: seven particulate matter species and five gaseous species over a span of ten years [20]. The pollutants chosen were those linked to adverse health effects and that which are currently measured by the United States Environmental Protection Agency (USEPA) [20,36]. These pollutants are PM_2.5_, particulate matter with an aerodynamic diameter less than or equal to 10 microns (PM_10_), nitrate (NO_3_^−^), ammonium (NH_4_^+^), sulfate (SO_4_^2−^), elemental carbon (EC), organic carbon (OC), carbon monoxide (CO), nitrogen dioxide (NO_2_), nitrogen oxides (NO_x_), sulfur dioxide (SO_2_), and ozone (O_3_). Cross validation by data withholding is then used to evaluate the DF method. This evaluation requires comparing the concentrations produced through the DF method to the available measurement values.

## 2. Materials and Methods

### 2.1. CMAQ Source

The *CMAQ* runs were performed by the USEPA using version 5.0.2 of *CMAQ* at a 12 km resolution across the contiguous United States for the years 2005–2014, for 12 pollutants of interest, to produce an hourly simulation [37,38]. The pollutants modeled were 7 particulate species (PM_10_, PM_2.5_, EC, OC, NH_4_^+^, SO_4_^2−^, NO_3_^−^) and 5 gases (CO, NO, NOx, O_3_, SO_2_). *CMAQ* relies on process-based principals to predict concentrations of species while taking into account emissions, meteorology, and chemical/physical properties [10,11]. *CMAQ* was run with the bidirectional NH_3_ air–surface exchange and the Carbon Bond version 2005 (CB05) chemical mechanism [37,38]. The simulations were run at a 12 km × 12 km resolution with 35 vertical layers that span till the top of the free troposphere, with layer 1 nominally 19 m tall [37]. Emission inputs were based on 2005, 2008, and 2011 National Emission Inventory (NEI) using the Sparse Matrix Operator Kernel version 3.1 [37,39]. A year-specific continuous emission monitoring system for large combustion and industrial process was also included [37,39]. A more detailed description of the input information is available through the USEPA (Appendix A). The surface level simulations were used in this DF method.

At a 12 km resolution, the contiguous United States is represented by a 299 × 459 cell matrix, with each cell centroid described through a latitude and longitude coordinate. The *CMAQ* simulation outputs hourly concentrations which are then converted to 24 h averages for particulate matter species, 1-h max average for gaseous species, and 8-h max average for ozone. These averaging methods were chosen based on the averaging methods of the current Environmental Protection Agency (EPA) standards of these pollutants [40,41]. Hourly concentrations from *CMAQ* at Greenwich Mean Time (GMT) were converted to the respective United States local time zone. The averaging was then performed from midnight to midnight local time. The time zone of each grid cell was determined through the latitude and longitude coordinates of the grid centroid.

### 2.2. Monitoring Data

The observational data was taken from the EPA’s Air Quality System (AQS) database for the same 2005–2014 time span and 12 pollutants [42]. This database contains observational air quality monitoring data from state, local, and tribal air pollution monitors across the contiguous United States [42]. Species such as ozone and PM_2.5_ had the best spatial coverage, while speciated PM had limited spatial coverage that was limited to metropolitan areas. Figure 1 shows a comparison between the spatial coverage of PM_2.5_ and EC across the contiguous United States.

The particulate matter species monitors sampled at three different frequencies: 1 in 6 days, 1 in 3 days, and daily. Monitors with daily measurements are only available for PM_2.5_ and PM_10_. The speciated PM (EC, OC, SO_4_^2−^, NO_3_^−^, NH_4_^+^) have monitors sampling 1 in 6 and 1 in 3 days. Table 1 shows the summary statistics for each of the species. The AQS PM data were downloaded as 24-hr averaged concentrations, the gaseous species (excluding ozone) were downloaded as 1-h max concentrations, and ozone was downloaded as 8-h max concentrations [42].

### 2.3. Data Fusion

To compensate for the monitoring data that are spatially and temporally incomplete, we fused the monitoring data to the *CMAQ* simulations. These simulations are independent of the monitoring data, based on emission estimates, meteorological data, and physical/chemical process modeling [10,11]. The first step is estimating two regression parameters α and β to correct the *CMAQ* simulation (Equation (1)).
(1)$$CMAQ_{corrected}=\alpha CMAQ^{\beta }$$

To estimate these parameters, we fit the above power regression model using annual averaged *CMAQ* ($\bar{CMAQ})$ as predictors of annual averaged observational $\left(\bar{OBS}\right)$ data. The annual averaged *CMAQ* values were taken at the cell locations corresponding to the observational data and only when observational data points were available at each location (Equations (2) and (3)). This is of importance for non-daily or seasonal monitors that only take measurements for certain portions of the year.

(2)$$\bar{OBS}=\frac{1}{n}\sum_{i\;\in m}OBS_{i}$$

(3)$$\bar{CMAQ}=\frac{1}{n}\sum_{i\;\in m}CMAQ_{i}$$

Here, *n* denotes the total number of observations taken at a point, *m* represents the days in the year in which the monitor takes a measurement, the overbar represents the annual average, and the subscript *i* represents the particular day. The β coefficient is limited between 0 and 1 to prevent any exponential increasing functions. The α and β coefficients are calculated for each year of each species to account for yearly changes between the *CMAQ* and observational data. The regression parameters are then used to adjust all *CMAQ* values within the matrix. Some scatter is expected in this model because of the spatial misalignment of the point source monitor and the gridded *CMAQ* field. The next step is to convert the measurements to a normalized dimensionless ratio (Equation (4)); this is the term that is spatially interpolated.

(4)$$\frac{OBS}{\bar{OBS}}\cdot \frac{\alpha {\bar{CMAQ}}^{\beta }}{\alpha CMAQ^{\beta }}$$

The α and β terms represent the regression parameters found through the power regression (Equation (1)), OBS is the daily observational value, $\bar{OBS}$ is the annual averaged observational data, CMAQ is the value at a cell with an observation, and $\bar{CMAQ}$ is the annual averaged simulated value at that cell. Rather than only normalizing by the average of the observations, the inclusion of the normalized *CMAQ* term accounts for monitors with only a partial year set of measurements. The *CMAQ* values are averaged only when observations are available. If multiple observations are included in one *CMAQ* grid cell, the average of the observations is used. For species such as elemental and organic carbon which only record observational values every 1 in 3 or 1 in 6 days, we used linear interpolation temporally to calculate the normalized dimensionless ratio on days with no observational data. This allows us to use the existing observational data to estimate the concentrations on days without any data available. Table 2 shows the range of α and β values calculated for each species over the ten-year period.

The α and β were calculated for each year to allow for variations in the relationship between the yearly *CMAQ* results and observational data, rather than finding one set of parameters to describe all ten years.

This normalized ratio is calculated at each monitoring location. These point values are then spatially interpolated using IDW to create a smooth spatial field of normalized dimensionless ratios to accommodate for areas with no observational data. The ratio of the IDW value was limited between 0.1 to 10 to prevent the spatial interpolation of any outlier *CMAQ* value that was over one order of magnitude larger or smaller than the measured value. This field is then multiplied with the adjusted field of *CMAQ* values to create the final fused field *C** (Equation (5)).
(5)$$C^{*}={\left(\frac{OBS}{\bar{OBS}}\cdot \frac{\alpha {\bar{CMAQ}}^{\beta }}{\alpha CMAQ^{\beta }}\right)}_{IDW}\times \alpha CMAQ^{\beta }$$

This method has fewer steps than previous data fusion methods that allows for easier implementation and less computational demands [20,21]. Equation (5) is applied daily to create a new fused field every day.

### 2.4. Model Performance Methods

Pearson squared (*R*^2^) was calculated temporally and spatially against the observational data to characterize method performance. The data for the temporal analysis is found by comparing the daily observational data at a monitor to the corresponding fused and *CMAQ* data values at the same location as the monitor. This is repeated for every monitor available for a particular pollutant in a given year. The spatial *R*^2^ is found by taking the correlation every day between the given observations on a particular day and the corresponding fused and *CMAQ* values (Equations (6) and (7)).

(6)$$temporal\;R^{2}=(corr{\left(OBS_{m}\left(t\right),\;modeled_{m}\left(t\right)\right)}^{2}$$

(7)$$spatial\;R^{2}={\left(corr\left(OBS_{t},\;modeled_{t}\right)\right)}^{2}$$

Here *m* represents the monitor and *t* represents the day.

### 2.5. Spatial Plots

To visualize the concentrations spatially, each pollutant is plotted geospatially using the *CMAQ* cell centroid coordinate information (latitude/longitude). Each grid cell is averaged over the ten-year time span and normalized to the maximum average concentration during the ten-year time frame (Equations (8) and (9)).
(8)$$C_{norm}^{*}=\frac{\bar{C^{*}}}{\bar{C_{max}^{*}}}$$
(9)$$\bar{C^{*}}=\frac{1}{n}\overset{n}{\underset{i=1}{\sum }}C^{*}$$

Here *C^*^* represents the fused concentration, the overbar represents the average over the ten-year time period, and *n* represents the total number of days in the domain.

### 2.6. Model Evaluation Methods

Random data withholding is used to evaluate the performance of the DF method. Ten groups of observational data are created, each with a unique 10% of the total monitoring data. The monitor data withheld in each set was chosen at random to eliminate spatial biases. The above data fusion method is performed with the remaining 90% of data for each of the ten groups, resulting in ten new fused fields. We then did our model performance measures on the new field at the removed measurement locations of that particulate group. All reported withholding results are the average of each of the ten runs. Ten runs were chosen to ensure the stability of the evaluation. The spatiotemporal *R*^2^ and normalized root mean squared error (NRMSE) were calculated at withheld data points to evaluate the model (Equations (10) and (11)).
(10)$$R^{2}={\left(corr\left(OBS,\;C^{*}\right)\right)}^{2}$$
(11)$$NRMSE=\frac{\sqrt{\frac{1}{N}\sum_{i=1}^{N}{\left(C_{i}^{*}\;-\;OBS_{i}\right)}^{2}}}{\bar{OBS}}$$

The *OBS* term represents all withheld observation points, *C** represents the fused field at those corresponding observation points, *N* represents the total number of withheld points, and the overbar represents the averaging of all observation points.

### 2.7. Population Weighted Average Exposure

Population weighted average exposure was calculated using the results from the fused fields over the ten-year domain for all species to understand long-term trends in air pollution. A fractional gridded population was calculated by determining the population in each 12 km *CMAQ* grid cell (Equation (12)).
(12)$$F_{pop}=\frac{P_{grid}}{P_{T}}$$

Here *F_pop_* represents the fractional value of population in a grid cell, *P_grid_* represents the population in the grid cell, and *P_T_* represents the total population in the contiguous United States. The population data are taken from the publicly available US Census Bureau. The 2010 census data was used for each year [43]. The tract data contains each tract’s population, the tract area, and coordinates for the population centroid [43]. Population in each cell was estimated by assigning cells to tracts and dividing the population of the tract equally among all of its assigned *CMAQ* grid cells. A cell was assigned to a tract if the centroid of a given cell fell within that tract. The fractional grid level population is then multiplied to the fused concentration and summed over the contiguous United States to determine the population weighted averaged exposure (Equation (13)).
(13)$$Population\;Weighted\;Exposure=\;\overset{n}{\underset{i=1}{\sum }}\left(F_{pop,\;i}*C_{i}^{*}\right)$$

The subscript *i* references the individual grid cell, *n* represents the total number of grid cells over the contiguous United States, and *C^*^* represents the calculated fused concentration.

## 3. Results

Four sets of results are presented. The first are normalized dimensionless plots of the resulting fused fields. Second, is the comparison of the performance metrics and the improvements made to the fused fields from the initial *CMAQ* fields. Third, the model evaluation results from the 10% data withholding and finally the results from the population weighted exposure.

Figure 2 The normalized fused fields averaged over the ten-year time span, normalized according to Equation (8).

The fused normalized fields varied across the 12 pollutants, which is expected because of the different nature of each species. Mobile source pollutants such as NOx, NO_2_, and CO were found to highlight metropolitan areas. Ozone was found to have the highest normalized values with high simulated concentrations over the Atlantic Ocean and Gulf of Mexico. SO_2_, a primary pollutant, was found to have a less homogeneous spatial distribution with high concentrations in the eastern contiguous United States area. Particulate species were found to have higher values in the eastern contiguous United States compared to the western side.

Model performance (Figure 3) was found to vary between species (Equations (6) and (7)). Pollutant estimates show an improved temporal and spatial *R*^2^ over the initial *CMAQ* fields to varying degrees. The NRMSE was highest in species that produced the most scatter in the initial *CMAQ* and observational data regression, such as PM_10_ and SO_2_ (Appendix A). The scatter is quantified in Appendix A where the *R*^2^ of the annual average *CMAQ* and observational regression is calculated.

All species showed an increase in performance for all metrics compared to the initial *CMAQ* results. O_3_, PM_10_, and SO_2_ were found to have the greatest increase in performance temporally, with SO_2_ having the lowest *CMAQ* and fused *R*^2^. These species (PM_10_ and SO_2_) had an increase in temporal *R*^2^ of 0.70. Temporal performance metrics were found to be higher than spatial metrics for all 12 species. The highest temporal *R*^2^ was seen for ozone at 0.98, with the lowest for SO_2_ at 0.84. Spatially, SO_4_^2−^ performed the best at 0.91, with SO_2_ as the lowest with 0.42. The greatest increase in spatial performance occurred for OC with an improved spatial *R*^2^ of 0.48.

Temporal performance was further examined by comparing the fused field to the observations on the different types of monitor sampling days. This is an additional way to quantify the effects of the different number of temporal observations available throughout the year. This data fusion method does not change based on the sampling frequency of the monitor, and accounts for monitors with both high and low sampling frequencies. Three pollutants are highlighted for this analysis: EC, SO_4_^2−^, and PM_2.5_. EC was chosen to highlight the effects of a sparsely populated monitoring network, SO_4_^2−^ was chosen to show a secondary pollutant, and PM_2.5_ was shown to highlight a highly populated monitoring network. Speciated PM monitors have two sampling frequencies: 1 in 3 days and 1 in 6 days. PM_2.5_ had the same two sampling frequencies, with additional daily sampling monitors. Table 3 compares the *R*^2^ and NRMSE on the different sampling frequencies. Day A represents the one in three sampling frequency, Day B represents the 1 in 6 sampling frequency, and Day C represents the daily sampling frequency for PM_2.5_. For speciated PM, Day A would have the fewest observational points because only the 1 in 3 day monitors are reporting a value. Day B would have the most observational points because both the 1 in 3 day monitors and 1 in 6 day monitors are reporting a value. For PM_2.5_ Day C would have the fewest observational points because only the daily monitors are reporting a value.

For the ten years, EC on average had an increased *R*^2^ of 0.02 and a decreased NRMSE of 0.02 on Day B compared to Day A. SO_4_^2−^ on average was not found to have a difference between Day A and Day B for *R*^2^ and NRMSE. PM_2.5_ was found to have decreased *R*^2^ and increased NRMSE for Day C compared to A and B. For PM_2.5_ and EC we see that performance is enhanced on days with more observational data.

A model evaluation was done by performing a 10% data withholding and rerunning the model to predict the fused concentration at the missing data points. The *R*^2^ and NRMSE for the data withholding are shown in Figure 4.

Ozone had the best evaluation metrics with the highest *R*^2^ of 0.86 and NRMSE of 0.14. All species had *R*^2^ and NRMSE with an improved withholding value compared to the original *CMAQ* model results. SO_2_ performed the worst with the lowest *R*^2^ of 0.21 and an NRMSE of 1.03. The *R*^2^ was found to increase for all species compared to the original modeled results on average of 0.22, with PM_2.5_ having the greatest increase of 0.35. The NRMSE was found to decrease for the withheld fields on an average of 0.14. NO_2_ was found to have the largest decrease of 0.36.

The evaluation metrics were split up between the western and eastern contiguous United States to further evaluate model spatial performance. Table 4 gives the specific eastern and western evaluation *R*^2^ and NRMSE for PM_2.5_, SO_4_^2−^, and EC. The western contiguous United States is defined as a monitor which falls west of the longitude line −94.6046. The metrics shown are the average of all the monitors within the respective domain for each particular year.

On average the eastern contiguous United States was found to have an increased *R*^2^ compared to the western contiguous United States of 0.14 and 0.18 for PM_2.5_ and SO_4_^2−^, respectively. This difference is found to be less pronounced in more recent years (2014) compared to earlier years. EC on average had a smaller difference in *R*^2^ of 0.02. PM_2.5_ and SO_4_^2−^ had an average decrease in NRMSE of 0.15 and 0.10, respectively, with EC having an average decrease in NRMSE of 0.08.

The fused results were then used to calculate a population weighted concentration across the contiguous United States (Equation (13)) and shown in Figure 5. This population weighted concentration was calculated for each CONUS covering grid cell and averaged nationally to determine an averaged contiguous United States population weighted exposure.

Pollutants such as SO_2_ and NO_x_ were found to have the largest decrease rates in concentration with a slope of −4.8 × 10^−3^ and −1.8 × 10^−3^, respectively (Appendix A). The slopes were calculated using linear regression for each species. All pollutants show a decrease in concentration for the time period as seen with the negative slopes (Appendix A). The oscillating peaks and valleys represent seasonal trends, which are more pronounced for species like O_3_ that have known seasonal patterns, and less pronounced for species like OC and NH_4_^+^ [10].

## 4. Discussion

The results from this data fusion method have the advantage of keeping the original spatial patterns and completeness of the *CMAQ* results, while also matching the temporal variability and correlation to the monitoring data. Yearly linear regression between the annual averaged *CMAQ* and observations (Equation (1)) was found to increase the variability regression parameters each year, in comparison to power regression. IDW was chosen over other spatial interpolation methods, such as ordinary kriging, because it provided greater stability with our daily spatial interpolations and was computationally less demanding.

Although the annual mean model results in Appendix A show scatter between the original *CMAQ* and observational data, a perfect 1:1 relationship is also not encouraged. Simulation CTM concentrations are an average concentration over a 12 km × 12 km grid, while the observational data represents a point measurement. Due to the wide nature of what falls within the *CMAQ* grid cell (highways, buildings) we should not expect the *CMAQ* to identically match the point source observational data. The goal of the annual mean model is not to match the *CMAQ* to the observational data identically, rather to capture the overarching trend of the observational data.

This method increased the temporal and spatial performance for all 12 species compared to the initial *CMAQ* field in both the performance and the evaluation. For the purposes of epidemiology and health studies temporal accuracy is important, making these results useful for these purposes. In comparison to solely using observational data, this method provides information in locations where observational data are sparse. This is particularly an issue for speciated particulate matter such as EC, OC, NO_3_, NH_4_^+^, and SO_4_^2−^ and for all species in rural areas.

A second advantage of this data fusion method is the scalability over a large domain (contiguous United States) and a wide range of species, without the need for calculating additional spatial and seasonal correction parameters. The ability to apply this method over the contiguous United States makes it computationally efficient, and easy to apply to other study domains, CTM field resolutions, and species. Current fusion methods with satellite data are limited primarily to gases and PM_2.5_ and do not deal with speciated PM on such a large domain. Observational data are available for a wider range of species compared to satellite data, allowing this method to improve model predictions for a wider range of species.

Furthermore, performing this fusion method over a ten-year domain allows us to visualize and quantify the rate of change in a species concentration over time (Figure 5). Decreases seen in mobile source pollutants such as CO, NO_2_, and NO_X_ reflect the improved car engine performance, control strategies, and increased regulation (Appendix A). The decrease seen in SO_2_ reflects the control strategies used at coal fired power plants (Appendix A). However, this method of data visualization is limited by the assumption of equally-distributed population weighted exposure within a grid cell.

This method can be limited by *CMAQ*’s performance in modeling certain species. Table 2 demonstrates that *CMAQ* has varying levels of performance in predicting different species, as indicated by the differing levels of *R*^2^ in the annual mean regression between the *CMAQ* and observations. For species such as PM_10_ with a lower *R*^2^ of 0.03–0.14, this can be due to limitations in understanding certain atmospheric and physical properties related to PM_10_. When *CMAQ* has limitations in modeling a species, the fusion method will also be limited in the improvements it can make. This can be seen in PM_10_ by the fused *R*^2^ being lower than other species like SO_4_^2−^, which *CMAQ* originally predicts well (*R*^2^ of 0.77–0.93).

A second potential limitation comes from the availability of monitoring data [10,11,12,15]. Although monitoring data are accurate, the spatial availability for speciated PM is lower compared to gases. In Table 1 we see than the monitors available for the speciated PM is an order of magnitude less than those available for gases such as ozone. This is evident in the spatial *R*^2^ performance. Every species showed an improvement of at least 50% from the original *CMAQ* data, however those with fewer monitors, EC and OC, showed a lesser improvement compared to species with a higher number of monitors (PM_10_, PM_2.5_, O_3_).

A monitoring network not included in this study was the Interagency Monitoring of Protected Visual Environments (IMPROVE) network which includes speciated PM_2.5_ concentrations in class 1 visibility areas throughout the contiguous United States [44]. The greatest improvement in including this dataset would likely occur in rural locations which previously had sparse observational data. Specifically, improvements would be greatest for species such as EC and OC in the western contiguous United States, where the Chemical Speciation Network reports few measurements and *CMAQ* does not predict the species as well as others (Table 2). There will also be an improvement to a lesser degree in the ion species as well (NO_3_^−^, NH_4_^+^, SO_4_^2−^). The ion species are more homogenous across the contiguous United States and are better predicted by *CMAQ*, compared to EC and OC. The IMPROVE network was not included in this study as an oversight, however the methodology to incorporate IMPROVE and any future observational datasets would not change.

## 5. Conclusions

The method described in this paper led to improved spatiotemporal exposure fields for 12 pollutants. This method is more readily implemented than previous data fusion methods. We have shown that this method is able to improve the temporal and spatial correlations compared to the initial *CMAQ* fields. Furthermore, this method has the potential to be applied to a wider range of species and a different domain, provided observational data are available.

## Figures and Tables

**Figure 1 ijerph-16-03314-f001:**
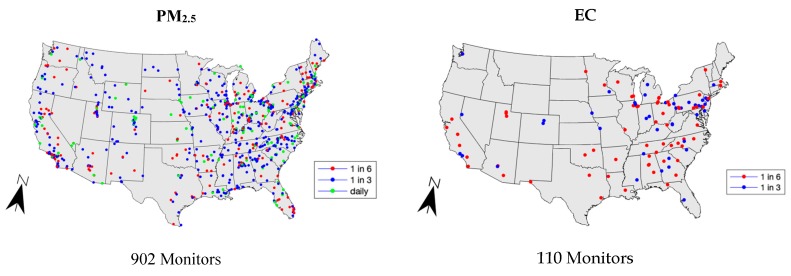
Comparison of monitor coverage between particulate matter (PM_2.5_) and elemental carbon (EC) for the year 2011. The different colored dots represent the different temporal sampling frequencies present for particulate species.

**Figure 2 ijerph-16-03314-f002:**
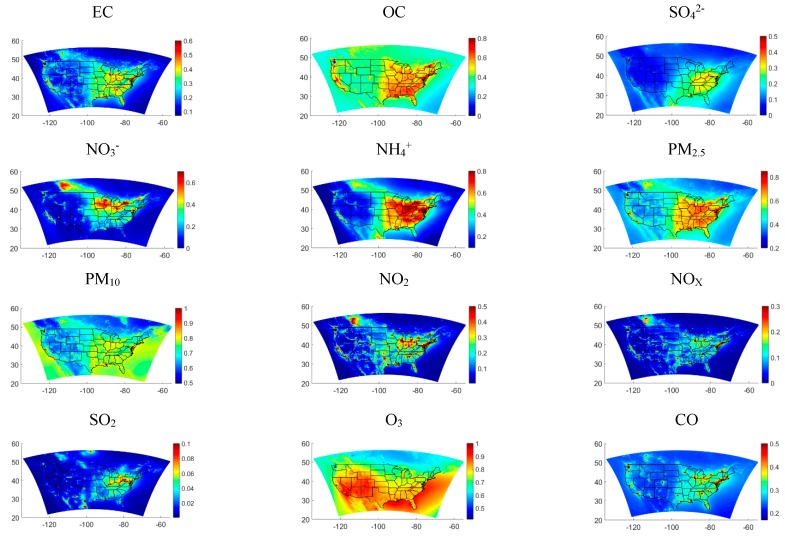
Normalized concentrations for the 12 species averaged over the ten-year time span. The concentrations are normalized against the maximum value over the contiguous United States (CONUS), excluding Mexico and Canada. The values plotted are dimensionless.

**Figure 3 ijerph-16-03314-f003:**
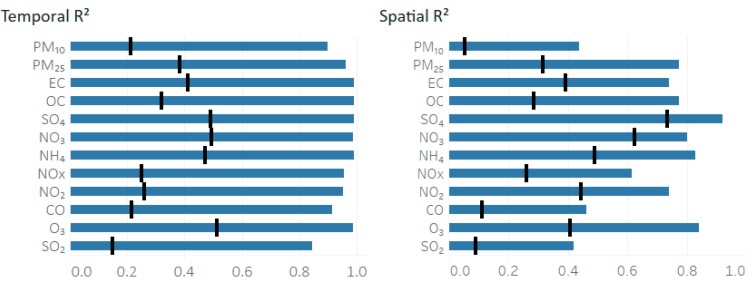
Comparison of fused field metrics (Pearson *R*^2^) to the original *CMAQ* model. The blue bar represents the fused field metric, while the black line for each species represents the *CMAQ* metric.

**Figure 4 ijerph-16-03314-f004:**
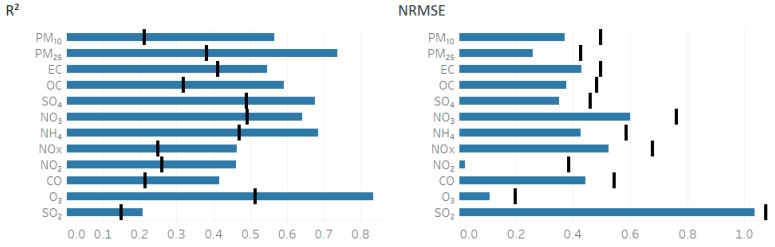
*R*^2^ and NRMSE used to evaluate model performance on the withheld dataset. The plots represent the average over each of the 10 withholding runs that were performed. The black line for each species represents the metric for the raw *CMAQ* model, while the blue bar represents the calculated withheld metric of the fused field.

**Figure 5 ijerph-16-03314-f005:**
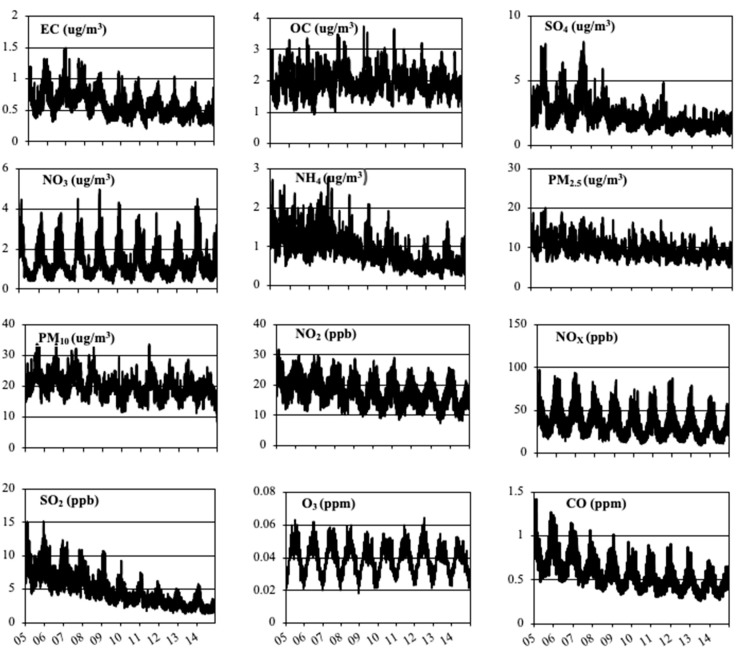
Nationwide averaged population weighted ambient concentrations plotted over the ten-year domain of 2005–2014 for the twelve pollutants. The *y*-axis shows the population weighted concentration with the units shown in the title, and the *x*-axis sows the dates ranging from 1 January 2005–31 December 2014.

**Table 1 ijerph-16-03314-t001:** Summary statistics for the observational data from 2005–2014. A range is presented to show the changes in observational data over the ten-year span. Completeness is defined as the percentage of days during the year with measurement data available. Monitor total represents the total number of active monitors during the year, while daily represents the number of active monitors that take daily measurements during the year. Total observations (OBS) represents the total measurements taken during the ten-year span.

	Pollutant	Monitor Total	Daily	Total OBS	Completeness (%)
Particulate Species	PM_10_	691–999	248–320	1,416,226	37–55
	PM_2.5_	768–1071	114–183	1,231,795	35–39
	EC	95–172	0	87,776	13–22
	OC	95–172	0	85,734	13–22
	SO_4_^2−^	103–172	0	102,311	20–23
	NO_3_^−^	103–172	0	90,176	20–23
	NH_4_^+^	102–172	0	94,332	19–22
Gases	NOx	308–419	268–371	1,164,912	85–90
	NO_2_	300–400	270–389	1,373,569	87–90
	CO	303–418	278–393	1,257,734	87–90
	O_3_	1182–1265	556–740	3,439,169	75–80
	SO_2_	429–507	396–481	1,572,601	90–93

**Table 2 ijerph-16-03314-t002:** The calculated parameter range for the α and β parameters and the range of the *R*^2^ for the fit between the community multiscale air quality (*CMAQ*) and observations. Each range encompasses the values used during the ten-year period.

		α	β	R^2^
Particulate Species	PM_10_	11.67–15.64	0.10–0.23	0.03–0.14
	PM_2.5_	3.68–4.62	0.32–0.50	0.28–0.50
	EC	0.58–0.84	0.34–0.54	0.26–0.49
	OC	1.45–1.80	0.18–0.45	0.10–0.35
	SO_4_^2−^	1.11–1.31	0.82–1.01	0.77–0.93
	NO_3_^−^	0.88–1.29	0.53–0.89	0.43–0.60
	NH_4_^+^	0.74–1.45	0.46–0.76	0.24–0.78
Gases	NOx	2.04–4.20	0.62–0.94	0.48–0.65
	NO_2_	1.66–2.21	0.68–0.76	0.63–0.71
	CO	0.82–1.09	0.43–0.67	0.16–0.36
	O_3_	0.32–0.73	0.64–0.91	0.50–0.65
	SO_2_	1.36–3.55	0.69–0.95	0.37–0.54

**Table 3 ijerph-16-03314-t003:** Comparison of the *R*^2^ and normalized root mean squared error (NRMSE) between C* and observations on the different sampling days. Speciated PM monitors, EC and SO_4_^2−^ have two sampling frequencies, and PM_2.5_ has three sampling frequencies. Day A represents 1 in 3 days, Day B represents 1 in 6 days, and Day C represents daily measurements (only for PM_2.5_). The metrics shown are the averages of all days included in each of the three categories.

		**2005**		**2006**		**2007**		**2008**		**2009**	
		***R*^2^**	**NRMSE**	***R*^2^**	**NRMSE**	***R*^2^**	**NRMSE**	***R*^2^**	**NRMSE**	***R*^2^**	**NRMSE**
EC	1 in 3	0.57	0.52	0.61	0.44	0.61	0.48	0.64	0.45	0.76	0.31
	1 in 6	0.59	0.49	0.62	0.42	0.64	0.44	0.68	0.38	0.76	0.31
SO_4_^2−^	1 in 3	0.90	0.24	0.91	0.52	0.92	0.18	0.87	0.23	0.92	0.20
	1 in 6	0.90	0.25	0.90	0.54	0.93	0.18	0.89	0.23	0.91	0.21
PM_2.5_	1 in 3	0.81	0.23	0.80	0.23	0.82	0.23	0.81	0.22	0.81	0.22
	1 in 6	0.81	0.24	0.80	0.24	0.81	0.23	0.81	0.22	0.80	0.22
	daily	0.77	0.26	0.76	0.25	0.80	0.25	0.80	0.27	0.80	0.30
		**2010**		**2011**		**2012**		**2013**		**2014**	
		***R*^2^**	**NRMSE**	***R*^2^**	**NRMSE**	***R*^2^**	**NRMSE**	***R*^2^**	**NRMSE**	***R*^2^**	**NRMSE**
EC	1 in 3	0.70	0.38	0.71	0.34	0.69	0.37	0.76	0.34	0.69	0.36
	1 in 6	0.68	0.41	0.72	0.32	0.72	0.31	0.76	0.33	0.71	0.33
SO_4_^2−^	1 in 3	0.85	0.31	0.91	0.19	0.90	0.21	0.89	0.20	0.87	0.24
	1 in 6	0.84	0.30	0.91	0.18	0.89	0.21	0.90	0.20	0.88	0.24
PM_2.5_	1 in 3	0.79	0.23	0.81	0.22	0.74	0.25	0.77	0.26	0.73	0.27
	1 in 6	0.78	0.23	0.81	0.22	0.74	0.26	0.76	0.26	0.74	0.28
	daily	0.78	0.24	0.78	0.24	0.74	0.26	0.75	0.26	0.72	0.27

**Table 4 ijerph-16-03314-t004:** Model evaluation metrics, *R*^2^ and NRMSE divided spatially into the eastern and western contiguous United States for PM_2.5_, SO_4_^2−^, and EC. The metrics are shown for each year and are the average of the monitors in the domain.

		**2005**		**2006**		**2007**		**2008**		**2009**	
		***R*^2^**	**NRMSE**	***R*^2^**	**NRMSE**	***R*^2^**	**NRMSE**	***R*^2^**	**NRMSE**	***R*^2^**	**NRMSE**
EC	Eastern	0.51	0.42	0.56	0.40	0.38	0.52	0.45	0.47	0.67	0.36
	Western	0.47	0.54	0.48	0.51	0.43	0.59	0.37	0.58	0.62	0.43
SO_4_^2−^	Eastern	0.81	0.32	0.76	0.32	0.79	0.31	0.74	0.33	0.71	0.31
	Western	0.52	0.45	0.57	0.41	0.54	0.44	0.48	0.43	0.56	0.41
PM_2.5_	Eastern	0.75	0.24	0.76	0.23	0.82	0.23	0.80	0.23	0.79	0.23
	Western	0.77	0.36	0.56	0.48	0.65	0.37	0.64	0.36	0.63	0.36
		**2010**		**2011**		**2012**		**2013**		**2014**	
		***R*^2^**	**NRMSE**	***R*^2^**	**NRMSE**	***R*^2^**	**NRMSE**	***R*^2^**	**NRMSE**	***R*^2^**	**NRMSE**
EC	Eastern	0.64	0.40	0.59	0.37	0.57	0.39	0.59	0.39	0.53	0.40
	Western	0.59	0.45	0.56	0.48	0.58	0.45	0.63	0.44	0.61	0.44
SO_4_^2−^	Eastern	0.68	0.35	0.73	0.33	0.65	0.32	0.72	0.32	0.65	0.34
	Western	0.42	0.51	0.58	0.40	0.58	0.37	0.61	0.39	0.54	0.46
PM_2.5_	Eastern	0.80	0.23	0.78	0.24	0.75	0.23	0.79	0.23	0.76	0.24
	Western	0.60	0.38	0.65	0.37	0.63	0.37	0.64	0.38	0.60	0.41

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
