# Peer review of "Application of a Fusion Method for Gas and Particle Air Pollutants between Observational Data and Chemical Transport Model Simulations Over the Contiguous United States for 2005–2014"

_ijerph, 2019, doi:10.3390/ijerph16183314_

Round 1
Reviewer 1 Report
See attached document.

Author Response
We would like to first thank you for taking the time to read over this manuscript and give your feedback and comments. They were very insightful and provided helpful corrections. We would like to take the opportunity to respond to all of the comments and point out the changes made to the manuscript. We feel that these changes help the reader better understand and interpret the results presented in this study.
Comment: Is the inverse distance weighting performed spatially in Eqn. 4 only performed spatially? If so were all collocated OBS and CMAQ used or only a subset, or only a subset?
Response: Yes, the inverse distance weighting was only performed spatially and all collocated observations and CMAQ simulations were used. If there were more than one observation in a CMAQ cell, then the average of observations was used.
Comment: Are there ever days within the CONUS in which there are no collocated OBS and CMAQ? (e.g. this might happen seasonally with ozone or with sparely monitored speciated pollutants)?
Response: Yes, there were days when there were no observations, for PM2.5 species. Although ozone did have many monitors that only reported measurements seasonally, many others provided data year-round and were used.
Comment: The explanation of and would be more easily understood id the regression were presented like the following
Response: The and coefficients were generated using the power regression equation given rather than the linear log transformation. If the log equation were used, the regression would yield different numerical coefficients since the error minimized would be different. Therefore, the power regression equation is presented in the manuscript.
Comment: The authors say that a temporal linear interpolation was performed on the normalized dimensionless ratio (Equ. 3) on the days that did not have observed data. How is interpolation handled at the beginning of the end of the year.
Response: The normalized ratio that was interpolated was calculated using the nearest available day when observational values were available, regardless of year. This occurred for speciated PM2.5.
Comment: In the Temporal Analysis section, the monitoring temporal R2, temporal NRMSE, and the temporal bias are calculated for each monitoring station/year/pollutant, correct?
Response: This is correct for temporal R2. Temporal R2 in Fig 3 is the average over all monitors. Spatial R2 is the average over all days. NRMSE and normalized bias (Figures 4 and Table S2) are calculated over space and time. The text has been corrected and revised to reflect this.
Comment: I believe there may be some typographical errors in some of the equations presented. If I am mistaken please disregard this comment. Equ 3 (and by extension Equ 4) should be written In other works the averaging symbol (AKA, the “bar” symbol) should extend all the way across the numerator. Also it may be possible that the CMAQ component of the normalized dimensionless ratio may not be need to be adjusted at all because the CMAQ adjustment happens in the second factor Equ 4.
Response: The bar was not extended over the and symbols because they were not part of the average. Rather they were applied to the annual averaged CMAQ term.
Comment: I also believe that Equ 6. And Equ 7 may be written incorrectly. I believe Equ 6 should be written and Equ 7 should be written . I believe Equ 7. Should be named “temporal normalized mean bias” instead of simply “temporal bias”. In section 2.4 (line 178), the bias was defined as absolute deviation. This would imply that the bias equation has some sort of absolute symbol, which it does not.
Response: Equations 5, 6 have been edited to show the temporal and spatial R2. The temporal R2 is the correlation at a monitor over time, the spatial R2 is the correlation on a day with all given monitors. Two new equations have been added to show the spatiotemporal R2 and NRMSE for the withholding analysis (figure 4). This is shown in equations 10, 11.
Comment: As a whole, it seems like the only purpose of parts of the methods were to be able to display 10 years’ worth of data in one figure. Although this is a succinct way to synthesize a large amount of data, such large averaging times do not allow for interesting exploration of the method and results and limit the reader’s ability to understand where and when the method performs well or poorly. Examples of large averaging times can be seen in Equ. 8 and Equ. 9. | do not understand the utility of these two equations. This does not seem like the most valuable way to display data results as seen in Figure 2. | am confused as to how to interpret Figure 2. Nearly all the displayed values in Figure 2 are less than 1, which seems obvious because the average concentration was being scaled by its maximum. None of the subplots in Figure 2 really show the performance of this data fusion method.
Response: Yes, the purpose of figure 2 was to give the reader a visual representation of the overall results. This figure was not intended to show the performance of the model, rather to give the reader a general understanding of the distribution of the ten pollutants over the domain. Model performance and evaluation were shown in figures 3 and 4 respectively. Additional performance and evaluation metrics have been added with Tables 3 and 4. Scaling to the maximum in Figure 2 provides a similar scaling across pollutants, although we varied the scale to show the variation.
Comment: The explanation of Figure 2 was not easily ascertained from the figure itself (lines 247-255). Where are major roadways and cities highlighted? Where are power plants located? These are not clear from Figure 2 alone
Response: A 12 km resolution will be too coarse for major roadways to be seen very clearly, but large population centers and roadways can be seen in primary pollutants such as EC, NOx, and CO. Coal-fired power plants can be detected in the SO2 plot.
Comment: Section 2.9 seems to be missing chunks that would explain the method and there seems to be a lack of utility in having a population weighted exposure. The results from this section and Figure 5 could have been explained a lot more. Some of parts of Figure 5 are also misleading. Did the authors have a population file for every year of data? Did they only use one population file for the entire time period? If so, from which census? The authors only say the file is “publicly available” from the “US Census Bureau”, but never give a citation. Did they have to do any kind of processing to get population from a given geographical boundary (e.g., census tract) to a grid?
Response: An additional explanation has been provided in the revised manuscript, with the description about how the US Census data has been translated to the gridded CMAQ field (shown below). One population file, from the year 2010 was used for the time period. A citation has been added which provides the web address where the data is downloaded from.
Manuscript: “The 2010 census data was used for the complete time domain[1]. The tract data contains each tract’s population, the tract area, and the coordinates for the population centroid[1]. Population in each cell was estimated by assigning cells to tracts and dividing the population of the tract equally among all of its assigned CMAQ grid cells. Cells were assigned to a tract if the centroid of a given tract was within that cell.”
Comment: In the subplots in Figure 5, the authors do not acknowledge that decrease in some of pollutants are directly dependent on decreases of other pollutants (i.e. the speciated PM pollutants are subsets of PM2.5 and PM2.5 is a subset of PM10). How does incorporating a population component change results compared with only having some sort of areal average? What is added by having a population component? The text speaks about “decreases” versus “largest decreases” (lines 292-297). However, the authors should speak more precisely about these decreases: are the quantified relatively or absolutely?
Response: The purpose of adding the population weighted time trend was to showcase the ten years’ worth of time trends for the 12 species. This could have also been shown with an aerial average over the contiguous United States or for a particular grid cell. We chose a population weighted concentration because of the health applications of the fused fields. Table S4 has been added to the supplemental section to show the results of linear regression for the 12 pollutions with the population weighted exposure. This table quantifies more precisely the decreasing rates seen in the figure. The manuscript has been revised to describe the decreases seen more quantitatively using the results of the linear regression.
Comment: The authors were correct in comparing the data fusion method with CMAQ. However, they should have also compared it with using an IDW using observational data alone. That way, the authors (and readers) have an idea by how much the improvement in the data fusion is really coming from the observed data and the CMAQ estimates.
Response: Previous data fusion work has explored interpolating observational data alone[2]. In this study presented by Hu et. al a comparison is performed of multiple interpolation methods (include IDW) against a CMAQ data fusion method. The results show that the data fusion method is an improvement compared to solely inverse distance weighting the observations. For pollutants with more sparsely located monitors that tend to be located in metropolitan areas (e.g., CO) interpolating observations alone can lead to substantial overprediction in rural areas where few or no monitors are located. By normalizing the observations to its CMAQ value, we are interpolating a spatial field with less bias caused by bias in monitor locations (i.e., lack of monitors in rural areas).
Comment: This manuscript deals with daily data from all 12 pollutants explored. How were hourly PM2.5 data dealt with from AQS?
Response: 24-hour averaged PM2.5 data was downloaded from the AQS database, so no additional processing was required for the observational data. An additional note has been added in the manuscript to indicate this:
Manuscript: “The AQS PM data was downloaded as 24-hr averaged concentrations, the gaseous species (excluding ozone) was downloaded as 1-hr max concentrations, and ozone was downloaded as 8-hr max concentrations.”
Comment: I wish the authors has performed some sort of sensitivity analysis on the averaging parameters used in the methods. How would Figure 3 and Figure 4 change if a, B, OBS, and CMAQ were averaged to six months or two years?
Response: The and parameters were recalculated for each year to allow for yearly changes in the observational and CMAQ results. For pollutants with larger relative decreases over the ten years, such as SO2, there is an increased sensitivity in the parameters compared to a pollutant more consistent in concentration over the ten-years (O3). This is also evident in table S1 where the yearly parameters for and are presented. Since CMAQ uses annual updates in some of its input models (e.g., emissions), we feel that regressing the years separately makes most sense.
Comment: The explanation of a 10-fold cross-validation was a bit confusing. People in the data fusion community know what a 10-fold cross-validation is, so any kind of detailed explanation is probably not needed. However, the authors say, “each monitor has been removed at least once” (line 211). Monitors should be removed EXACTLY once. Typically, in a 10-fold cross-validation, monitoring locations are randomly assigned to 10 different groups and each group is removed once. In this manuscript, are groups based off monitoring locations or observed data regardless of location?
Response: The manuscript has been edited to address this point. Each group of monitors was removed exactly once with the fusion performed with the remaining data. The monitors were not chosen on their monitoring location.
Manuscript: “Ten groups of observational data are created, each with a unique 10% of the total monitoring data. The monitors withhold in each set was chosen at random to eliminate spatial biases.”
Comment: How does Figure 4 change spatially and temporally? The manuscript would benefit if the authors inserted a map of the change in R? or NRMSE. This could be done for either a few selected years and/or a few selected pollutants. There is no sense from the manuscript which locations in CONUS would most benefit the most from this data fusion method.
Response: This would be a great metric to add to the analysis. We have added Table 3 and 4 to the manuscript in the results section. These tables give a specific case for PM2.5, SO4, and EC to further examine temporal and spatial evaluation and performance. We have chosen to give a specific example for these pollutants to highlight the effect of a sparsely populated monitoring network, a secondary pollutant, and a densely populated monitoring network. Table 3 is designed to show the difference in performance on the different types of monitor sampling days. Table 4 is designed to show the performance specifically in the eastern and western contiguous United States.
Comment: The authors mentioned that a “three-step method was implemented over Georgia” (lines 174-175) but the authors never gave a citation, never explained this method, and never implemented it. | do not think that comparison method is necessarily needed for this manuscript, but this text should be removed if the authors are not going to implement or explain this “three-step method”.
Response: The references to the three-step method was removed in the referenced lines 174-175. Rather a note has been added about how this new method allows for easier implementation.
Manuscript: “This method has fewer steps than previous data fusion methods that allows for easier implementation and lesser computational demands[3, 4].”
Comment: The authors make a point that their method is only one step (lines 169-175). However, the text and language used by the authors contradict this statement. The authors use the language “the first step” (line 119) and “the next step” (line 137) when describing the method, implying that there is more than one step. As a reader, | saw that the method had five steps: 1) find and , 2) find OBS and CMAQ, 3) calculate Equ. 3, 4) calculate the temporal linear interpolation, and 5) calculate Equ. 4. For me, I do not think that the number of steps make a data fusion method better or worse, but rather the computational time and technical expertise needed to implement the data fusion method. The method developed in this manuscript is both computational efficient and straight-forward to implement. I would focus on that instead of the number of steps.
Response: The references to the “one-step” method have been removed from the manuscript to lessen confusion regarding the methodology. As addressed in the previous response a note has been added to the methods section that highlights the ease of implementation and lessened computational demands in comparison to previous fusion methods.
Comment: The limitations of the method were not sufficiently explained. IDW has its own problems that were never described. Namely, if measured values are clustered spatiotemporally, they are likely to be autocorrelated meaning they are likely contributing the same information, giving them an artificially greater weight than observed values further away. The method was also limited by predictions not having an autoregressive component.
Response: An additional note was added to the manuscript regarding why IDW was chosen over other spatial interpolation methods such as kriging. Kriging a domain this large would have significantly increased the computational demands for a domain of this size. Furthermore, when we implemented kriging, we found that the model has some instability in locations where monitors were clustered spatially in one location and predicting different values. This created negative weights which would imply a negative nonphysical concentration. This problem doesn’t arise in inverse distance weighting because the calculated value is a weighted average of the neighboring points.
Manuscript: “IDW was chosen over kriging because it provided greater stability with our daily interpolation, and was computationally less demanding. However there are known limitations with IDW regarding the interpolation domain size and the scarcity of data points[5].”
Comment: There are instances where PM2.5 is called PM25
Response: This has been corrected.
Comment: There are instances where pollutants have inconsistent subscripts (PM2.5 instead of PM2.5).
Response: This has been corrected
Comment: Consistent use throughout of using oxford commas or not.
Response: This has been corrected
Comment: Consistency in text between “US” versus “United States”
Response: This has been corrected
Comment: some acronyms were defined twice: “CMAQ” (line 72) and IDW(line 161).
Response: This has been corrected
Comment: Increase the resolution of Figure 1.
Response: This has been corrected
Comment: Is there a difference between and (line 125)?
Response: has been changed to
Comment: There were a few instances where a sentence started with either a number of symbol
Response: This has been corrected
Comment: There is a typo in Figure S1. Two subplots are labeled “SO4” and “NH4”
Response: This has been corrected
Comment: Is it possible to have the numbers presented in Figure 3 and 4 be put in a supplementary table?
Response: This has been added to the supplementary section (Table S2, S3).
Comment: Citations needed int eh following places: sentence on lines 81-83 and sentence on lines 319-320.
Response: This has been corrected
Comment: Last sentence in the Figure 1 caption did not need to be there.
Response: This has been corrected
Comment: Redundancy in certain phrases: “Speciated PM species”, “CTM models”, etc.
Response: This has been corrected
Comment: Odd capitalization of “Ozone” throughout
Response: This has been corrected
Comment: The authors sometimes say “United States” when they mean “contiguous United States”
Response: This has been corrected
Comment: References 34-35 are not formatted correctly.
Response: This has been corrected
References:
Bureau, U.S.C. Decennial Census of Population and Housing. Available from: https://www.census.gov/programs-surveys/decennial-census/data/tables.2010.html. Yu, H., et al., Cross-comparison and evaluation of air pollution field estimation methods. Atmospheric environment, 2018. 179: p. 49-60. Friberg, M.D., et al., Method for fusing observational data and chemical transport model simulations to estimate spatiotemporally resolved ambient air pollution. Environmental science & technology, 2016. 50(7): p. 3695-3705. Huang, R., et al., Air pollutant exposure field modeling using air quality model-data fusion methods and comparison with satellite AOD-derived fields: application over North Carolina, USA. Air Quality, Atmosphere & Health, 2018. 11(1): p. 11-22. Wong, D.W., L. Yuan, and S.A. Perlin, Comparison of spatial interpolation methods for the estimation of air quality data. Journal of Exposure Science and Environmental Epidemiology, 2004. 14(5): p. 404.

Reviewer 2 Report
This manuscript summarizes an ambitious project designed to create a comprehensive air quality database for exposure assessment in health studies for the entire contiguous U.S. for the ten-year period from 2005 through 2014. It fuses CMAQ model output results with observational data to fill in temporal and spatial gaps present in the existing observational data records. The most significant result in my opinion is Figure 5 which shows the nationwide averaged population weighted ambient concentrations of the 12 pollutants. The fused analysis is very well suited for this type of analysis. The data fusion also shows promise in filling in gaps in the temporal air quality record in metropolitan areas with relatively good data coverage. However, the CMAQ model output required significant corrections to remove biases and it is unclear whether the data fusion method is able to provide reasonable spatial variations in data sparse regions (such as the western US). I have made a suggestion on how this weakness can be addressed in #1 below.
Major Comments/Suggestions
1) The project is severely hampered by the paucity of PM speciation data, especially in the western US (See Fig 1). The maximum number of PM speciation stations available in any of the ten years was 172 (Table 1) with only 16 of these west of the Rocky Mountains. This project used PM speciation data obtained from the EPA’s Air Quality System (AQS) database (line 90). However, line 94 states that the “speciated PM species had limited spatial coverage that was limited to metropolitan areas” (See also Fig 1). This leads me to believe that this project relied exclusively upon data from the PM2.5 Chemical Speciation Network (CSN) and failed to include data from the Interagency Monitoring of Protected Visual Environments (IMPROVE) aerosol network. The IMPROVE network, which was initiated in the late 1980s has one of the longest, continuous aerosol chemical composition records in the world. Because the network was designed to monitor visibility in Class I Wilderness Areas, the sites are almost exclusively outside of metropolitan areas. In fact, there are more than 100 IMPROVE/IMPROVE Protocol sites in the western US available through the AQS database. In addition, the IMPROVE and CSN networks utilize similar sampling and analysis protocols so the data should be comparable. A description of the available speciated PM data from various US Networks can be found at file:///C:/Users/u0302380/Downloads/FINAL%20FINAL%20PAUL%20-%20MANUSCRIPT_FINAL_CLEAN_5-28-14%20WITH%20TABLE%20AND%20FIG.%20(1).PDF. I am assuming that this was an oversight and not intentional. If the IMPROVE data were intentionally excluded, then the text should justify the decision. If not, then I would recommend redoing the analysis including all of the IMPROVE and IMPROVE protocol sites. This would approximately double the number of available PM speciation stations and provide critical data in the western US.
2) The manuscript goes to great lengths to reduce biases in the CMAQ results using a power law correction (Equation 1). However, the text never summarizes the known biases determined from previous publications (see lines 86-87). The efficacy of the data fusion procedure is limited by ability of the CMAQ model to reproduce the basic spatial and temporal variations of the air pollutants. Thus, it is important to have more background on the strengths and weaknesses of the CMAQ model. In addition, no information is provided about the emission inventories used in the CMAQ simulations or the vertical resolution of the model. I assume that the emission inventory includes known point and mobile sources. However, does it include fugitive emissions from dust or wildfires? This would be a big problem in the western US where these sources are significant. In addition, the fitted alpha and beta parameters shown in Table 1 differ significantly from 1.0. This results in very aggressive corrections to the CMAQ model output and raises concerns about the overall performance of CMAQ. Lastly, no justification was provided as to why a power law regression model is the best choice to correct the CMAQ output.
3) Suggestion – I would really like to see the data from Figures 3 and 4 broken down for the eastern and western US separately in a Supplemental figure. I suspect that there are big differences in CMAQ and data fusion results in these two regions because the dominant aerosol species differ significantly.
Minor Comments/Suggestions
Line 21 – NO2 should be NO
Line 21 – Be consistent with the inclusion of the oxidation state of the PM species (i.e., NO3-, NH4+, etc.)
Line 39 – “emission of a single monitor” should be “emissions on a single monitor”
Line 45 – “pollutant field” should be plural
Line 56 – “data are” should be “data”
Line 57 – Spell out aerosol optical depth before using the AOD acronym
Lines 66 and 67 – “concentration” and “value” should be plural
Line 74 – “produce hourly simulation” should be “produce and hourly simulation”
Line 75 – formatting issue with PM2.5 subscript
Line 76 – states that the 12 pollutants were chosen for their known adverse health effects. However, the EPA only regulates 6 of the 12 as criteria air pollutants (which have documented health effects). Either provide documentation for the health impacts of the other 6 or alter the rationale.
Line 80 – 24-hr averages are 0 GMT to 0 GMT or midnight to midnight local? The text is not clear.
Line 93 – no need to capitalize ozone
Line 104 – include oxidation state for PM species for consistency
Table 1 – No need to capitalize statistics in caption
Table 1 – There seems to be a problem with the total OBS column for the PM speciated data. If we assume 172 monitors x 365 days x 10 years x 1 day in 3 frequency then the maximum number of observations would be 209,267. 1 day in 6 would be half of that. Since there is a mix of frequencies for the PM speciation data with none of them being daily the total number of OBS should be between 209267 and 104,633.
Table 1 – include oxidation state for PM species for consistency
Line 197 – “This is analysis” should be “This analysis”
Figure 2 – include oxidation state for PM species for consistency
Line 249-250 – Ozone does not appear elevated over the Pacific Ocean in Figure 2 as stated in the text. Ozone is elevated over the Atlantic and the Gulf of Mexico
Line 261 and 264 directly contradict each other with regards to which species had the greatest increase in performance temporally. I believe that line 264 is correct based upon Figure 3.
Line 276 – no need to capitalize ozone
Line 277 – I have no idea what this text is referring to
Line 278 – The stated R2 and NRMSE values of 0.48 and 0.42 appear to be for CO rather than SO2 as stated in the text. I believe the correct values should be 0.21 and ~1
Line 281-282 – “The bias decreased for all species other than NO3 and PM10” This statement is not supported by Figure 4 which shows a bias decrease for all species
Figure 5 – units on NO2 should be ppb
Table S1 – include oxidation state for PM species for consistency
Table S1 – labels for SO2, O3, and CO are incorrect. They should be ppb, ppm, and ppm.
Figure S1 – Units on CO should be ppm and not ppb
Author Response
We would like to first thank you for taking the time to read over this manuscript and give your feedback and comments. They were very insightful and provided helpful corrections. We would like to take the opportunity to respond to all of the comments and point out the changes made to the manuscript. We feel that these changes help the reader better understand and interpret the results presented in this study.
Comment: The project is severely hampered by the paucity of PM speciation data, especially in the western US (See Fig 1). The maximum number of PM speciation stations available in any of the ten years was 172 (Table 1) with only 16 of these west of the Rocky Mountains. This project used PM speciation data obtained from the EPA’s Air Quality System (AQS) database (line 90). However, line 94 states that the “speciated PM species had limited spatial coverage that was limited to metropolitan areas” (See also Fig 1). This leads me to believe that this project relied exclusively upon data from the PM2.5 Chemical Speciation Network (CSN) and failed to include data from the Interagency Monitoring of Protected Visual Environments (IMPROVE) aerosol network. The IMPROVE network, which was initiated in the late 1980s has one of the longest, continuous aerosol chemical composition records in the world. Because the network was designed to monitor visibility in Class I Wilderness Areas, the sites are almost exclusively outside of metropolitan areas. In fact, there are more than 100 IMPROVE/IMPROVE Protocol sites in the western US available through the AQS database. In addition, the IMPROVE and CSN networks utilize similar sampling and analysis protocols so the data should be comparable. A description of the available speciated PM data from various US Networks can be found at file:///C:/Users/u0302380/Downloads/FINAL%20FINAL%20PAUL%20-%20MANUSCRIPT_FINAL_CLEAN_5-28-14%20WITH%20TABLE%20AND%20FIG.%20(1).PDF. I am assuming that this was an oversight and not intentional. If the IMPROVE data were intentionally excluded, then the text should justify the decision. If not, then I would recommend redoing the analysis including all of the IMPROVE and IMPROVE protocol sites. This would approximately double the number of available PM speciation stations and provide critical data in the western US.
Response: Thank you for notifying us about the additional speciated monitors in the IMPROVE network. This was an oversight on our behalf and should have been included for the speciated PM monitors. However, given the additional time needed to incorporate the network into the fused fields, we do not believe there will be enough time to include this data into this current manuscript. However, to address your concerns of spatial performance in the western contiguous United States we have broken up the metrics in Figure 4 (R2 and NRMSE) to highlight the differences in the western and eastern contiguous United States for 3 pollutant cases: PM2.5, EC, and SO4. EC was selected to show a sparely populated monitoring network, PM2.5 was chosen to show a high populated monitoring network, and SO4 was chosen to highlight a secondary pollutant. The results of this analysis for the ten-year span is shown in Table 4.
Manuscript: “The evaluation metrics were split up between the western and eastern contiguous United States to further evaluate model spatial performance. Table 4 gives the specific eastern and western evaluation R2 and NRMSE for PM2.5, SO42-, and EC. The western contiguous United States is defined as a monitor which falls west of the latitude line -94.6046. The metrics shown are the average of all the monitors within the respective domain for each particular year”
Comment: The manuscript goes to great lengths to reduce biases in the CMAQ results using a power law correction (Equation 1). However, the text never summarizes the known biases determined from previous publications (see lines 86-87). The efficacy of the data fusion procedure is limited by ability of the CMAQ model to reproduce the basic spatial and temporal variations of the air pollutants. Thus, it is important to have more background on the strengths and weaknesses of the CMAQ model.
Response: The manuscript has added an additional section in the introduction to further detail the known biases found from the literature review.
Manuscript: “Examples of this bias include over predictions of particulate nitrate (NO3-) and ammonium (NH4+) and underpredictions of carbonaceous aerosols in spring and summer[1]. For ozone (O3) these biases include an underestimation in North America of daytime O3 in the spring and an overestimation of daytime concentrations during the summer[2, 3]. Evaluation of CMAQ to the Ozone Monitoring Instrument (OMI) found that CMAQ overestimates summertime nitrogen dioxide (NO2) in urban locations and underestimates in rural locations[4].”
Comment: In addition, no information is provided about the emission inventories used in the CMAQ simulations or the vertical resolution of the model. I assume that the emission inventory includes known point and mobile sources. However, does it include fugitive emissions from dust or wildfires? This would be a big problem in the western US where these sources are significant.
Response: The Emission inventories used by the USEPA came from the NEI emissions for the year 2005, 2008, and 2011. The model was run on a 12km x 12km resolution with 35 vertical layers. The vertical layers span the height of the troposphere with the height of each layer being 19m. These emission inventories include known point and mobile sources, as well as included emissions from fugitive dust emissions. An additional note has been added in the manuscript in section 2.1 CMAQ source to detail input parameters of the model. The emission inventory includes fugitive dust emissions and a yearly wildfires file. We have included a section in the supplemental material (Section S1) that links all the metadata used by the EPA to setup the CMAQ runs.
Manuscript: CMAQ was run with the bidirectional NH3 air-surface exchange and the Carbon Bond version 2005 (CB05) chemical mechanism[5]. The simulations were run at a 12km x 12km resolution with 35 vertical layers that span till the top of the free troposphere, each layer with a height of 19m[5]. Emission Inputs were based on 2005, 2008, and 2011 National Emission Inventory (NEI) using the Sparse Matrix Operator Kernel version 3.1[5, 6]. A year specific continuous emission monitoring system for large combustion and industrial process was also included[5, 6].
Comment: In addition, the fitted alpha and beta parameters shown in Table 1 differ significantly from 1.0. This results in very aggressive corrections to the CMAQ model output and raises concerns about the overall performance of CMAQ. Lastly, no justification was provided as to why a power law regression model is the best choice to correct the CMAQ output.
Response: Some level of scatter is expected between the CMAQ and observational data because of the spatial misalignment of the 12km x 12km averaged CMAQ grid cell and the point source observational data. Furthermore, it is possible than multiple monitoring measurements can fall into one CMAQ grid cell. The goal of the annual model is to not predict the observations exactly, but to predict a 12km x 12km average for the CMAQ cell. A power law regression was chosen over other regression methods, particular linear regression, because a power law regression provides conservative predictions for potentially high outlier CMAQ values. The and parameters are used to correct the daily CMAQ simulation outputs where occasional extreme high values can be predicted in certain grid cells. A linear regression would over correct the cell to a higher value than a power law regression model.
Comment: I would really like to see the data from Figures 3 and 4 broken down for the eastern and western US separately in a Supplemental figure. I suspect that there are big differences in CMAQ and data fusion results in these two regions because the dominant aerosol species differ significantly.
Response: This is a helpful way to understand the model performance. By the definition of the model performance metrics there would be very small differences between the eastern and western contiguous United States. The is because the model is designed to match the trends at the monitoring location. However, in model evaluation we can test the model’s ability to predict in the eastern and western US by withholding a specific monitor in the eastern and western contiguous United States. As mentioned in the first response, Table 4 was added to show the differences in model performance for the western and eastern contiguous United States. An additional table was added (Table 3) to show the difference in model temporal performance on the different monitor sampling day types. Speciated PM has two measurement day types: 1 in 3 and 1 in 6 days. Every 6 days all observational data will align, with the 1 in 6 and 1 in 3 day monitors both reporting a value. Every third day in-between only has a 1 in 3 day reporting a value. By evaluating the R2 and NRMSE on these two different types of days we can further temporally evaluate the model performance. These results are presented for all ten years for the same select three pollutants as Table 3.
Comment: states that the 12 pollutants were chosen for their known adverse health effects. However, the EPA only regulates 6 of the 12 as criteria air pollutants (which have documented health effects). Either provide documentation for the health impacts of the other 6 or alter the rationale.
Response: A reference has been added to quantify the negative health effects of speciated PM[7].
Comment: Line 80 – 24-hr averages are 0 GMT to 0 GMT or midnight to midnight local? The text is not clear.
Response: The averages were performed midnight to midnight to midnight local time. This has been clarified in the manuscript text.
Comment: Line 21 – Be consistent with the inclusion of the oxidation state of the PM species (i.e., NO3-, NH4+, etc.)
Response: This has been corrected
Comment: Line 39 – “emission of a single monitor” should be “emissions on a single monitor”
Response: This has been corrected
Comment: Line 45 – “pollutant field” should be plural
Response: This has been corrected
Comment: Line 56 – “data are” should be “data”
Response: This has been corrected
Comment: Line 57 – Spell out aerosol optical depth before using the AOD acronym
Response: This has been corrected
Comment: Lines 66 and 67 – “concentration” and “value” should be plural
Response: This has been corrected
Comment: Line 74 – “produce hourly simulation” should be “produce and hourly simulation”
Line 74 – “produce hourly simulation” should be “produce and hourly simulation”
Response: This has been corrected
Comment: Line 75 – formatting issue with PM2.5 subscript
Response: This has been corrected
Comment: Line 93 – no need to capitalize ozone
Response: This has been corrected
Comment: Line 104 – include oxidation state for PM species for consistency
Response: This has been corrected
Comment: Table 1 – No need to capitalize statistics in caption
Response: This has been corrected
Comment: – There seems to be a problem with the total OBS column for the PM speciated data. If we assume 172 monitors x 365 days x 10 years x 1 day in 3 frequency then the maximum number of observations would be 209,267. 1 day in 6 would be half of that. Since there is a mix of frequencies for the PM speciation data with none of them being daily the total number of OBS should be between 209267 and 104,633
Response: This has been corrected
Comment: Table 1 – include oxidation state for PM species for consistency
Response: This has been corrected
Comment: Line 197 – “This is analysis” should be “This analysis”
Response: This has been corrected
Comment: Figure 2 – include oxidation state for PM species for consistency
Response: This has been corrected
Comment: Line 249-250 – Ozone does not appear elevated over the Pacific Ocean in Figure 2 as stated in the text. Ozone is elevated over the Atlantic and the Gulf of Mexico
Response: This has been corrected
Comment: Line 261 and 264 directly contradict each other with regards to which species had the greatest increase in performance temporally. I believe that line 264 is correct based upon Figure 3
Response: This has been corrected
Comment: Line 276 – no need to capitalize ozone
Response: This has been corrected
Comment: Line 277 – I have no idea what this text is referring to
Response: This text has been corrected
Comment: Line 278 – The stated R2 and NRMSE values of 0.48 and 0.42 appear to be for CO rather than SO2 as stated in the text. I believe the correct values should be 0.21 and ~1
Response: This has been corrected, in addition table S2 has been added to the supplemental material to show the numerical values from figure 3.
Comment: Line 281-282 – “The bias decreased for all species other than NO3 and PM10” This statement is not supported by Figure 4 which shows a bias decrease for all species
Response: This text has been removed
Comment: Figure 5 – units on NO2 should be ppb
Response: This has been corrected
Comment: Table S1 – include oxidation state for PM species for consistency, labels for SO2, O3, and CO are incorrect. They should be ppb, ppm, and ppm.
Response: This has been corrected
Comment: Figure S1 – Units on CO should be ppm and not ppb
Response: This has been corrected
References:
Appel, K.W., et al., Evaluation of the community multiscale air quality (CMAQ) model version 4.5: sensitivities impacting model performance; part II—particulate matter. Atmospheric Environment, 2008. 42(24): p. 6057-6066. Appel, K.W., et al., Examination of the Community Multiscale Air Quality (CMAQ) model performance over the North American and European domains. Atmospheric Environment, 2012. 53: p. 142-155. Appel, K.W., et al., Evaluation of the Community Multiscale Air Quality (CMAQ) model version 4.5: sensitivities impacting model performance: part I—ozone. Atmospheric Environment, 2007. 41(40): p. 9603-9615. Canty, T., et al., Ozone and NOx chemistry in the eastern US: evaluation of CMAQ/CB05 with satellite (OMI) data. Atmos. Chem. Phys, 2015. 15(19): p. 10965-10982. Zhang, Y., et al., A measurement‐model fusion approach for improved wet deposition maps and trends. Journal of Geophysical Research: Atmospheres, 2019. Agency, U.E.P., National Emissions Inventory (NEI), Facility-level, US, 2005, 2008, 2011. Fuentes, M., et al., Spatial association between speciated fine particles and mortality. Biometrics, 2006. 62(3): p. 855-863.

Reviewer 3 Report
To keep the temporal and spatial information for air quality, this study tried to combine the temporal variations of the observational data with spatial distribution from chemical transport model (CTM) results by fusion method. The topic is significant to air pollution monitoring. Overall, the manuscript was organized with clear descriptions in background, objective, proposed experiment, and analysis. However, the principle of proposed approach and data usage are still weak for examining the objective of this topic, which are the major concerns about this work. I therefore don’t recommend the present study to be published in IJERPH. However, there are four suggestions and one question to better the completeness of this study. The followings are some of the comments on the manuscript.
1. The input/initial of CMAQ model should be provided, for example a table list of data used.
2. The daily fluctuations of pollutants in concentration are obvious as Table 1 showed, thus the finer temporal resolution of data fusion is suggested for accurate results.
3. The principle/mechanism to support the correction of CMAQ simulation (Eq. 1) is not clear and should be clarified. In addition, the annual amount correction (Eq. 2) will induce large uncertainty as the results in Table 2 and Table S1.
4. The English writing should be improved as well.
Author Response
We would like to first thank you for taking the time to read over this manuscript and give your feedback and comments. They were very insightful and provided helpful corrections. We would like to take the opportunity to respond to all of the comments and point out the changes made to the manuscript. We feel that these changes help the reader better understand and interpret the results presented in this study.
Comment: The input/initial of CMAQ model should be provided, for example a table list of data used.
Response: We have included an additional note in section 2.1 of the Material and Methods section to describe the model inputs/initials in further detail. We have also included a link in the supplemental section (Section S1) of the paper providing a more detailed model parameter input used by the USEPA in setting up the CMAQ simulations. A citation has been included to a paper that uses the same publicly available EPA CMAQ simulations that details the model inputs. In addition, the National Emissions Inventory (NEI) has also been included as a citation.
Manuscript: “CMAQ was run with the bidirectional NH3 air-surface exchange and the Carbon Bond version 2005 (CB05) chemical mechanism[1]. The simulations were run at a 12km x 12km resolution with 35 vertical layers that span till the top of the free troposphere, each layer with a height of 19m[1]. Emission Inputs were based on 2005, 2008, and 2011 National Emission Inventory (NEI) using the Sparse Matrix Operator Kernel version 3.1[1, 2]. A year specific continuous emission monitoring system for large combustion and industrial process was also included[1, 2].”
Comment: The daily fluctuations of pollutants in concentration are obvious as Table 1 showed, thus the finer temporal resolution of data fusion is suggested for accurate results.
Response: Table 1 shows the range of and over the ten years. Table S1 shows the individual and that was calculated for each year. Although one and parameters are calculated for the year, the DF method is applied daily, with the daily observational measurements and daily CMAQ simulations. The manuscript has been edited to emphasize this point.
Manuscript: “Eqn 4 is applied daily to create a new fused field every day.”
Comment: The principle/mechanism to support the correction of CMAQ simulation (Eq. 1) is not clear and should be clarified. In addition, the annual amount correction (Eq. 2) will induce large uncertainty as the results in Table 2 and Table S1.
Response: An additional note has been added in the introduction sections to further describe the known biases present in the CMAQ simulation. Our literature search has shown a level of biases present in all modeled outputs, with certain species having a larger relative bias than other species. Therefore, with these known biases we chose to implement Eqn. 1 to correct the CMAQ simulation prior to creating the fused fields. An additional section has been added to the manuscript to explain why power regression was chosen over a linear regression. A power regression was found to provide a more conservative estimation for any outlier high concentration produced by the CMAQ simulation. Furthermore, we highlight in the manuscript that some level of scatter is expected between the CMAQ simulations and the 12km observations. This scatter can occur for reasons like the spatial misalignment of the 12km grid cell average to a point source, or if multiple observational points are located in one grid cell. The annual mean model is not designed to predict the observational point sources exactly, rather it is designed to predict a corrected 12km averaged grid cell value.
Manuscript: “Examples of this bias include over predictions of particulate nitrate (NO3-) and ammonium (NH4+) and underpredictions of carbonaceous aerosols in spring and summer[3]. For ozone (O3) these biases include an underestimation in North America of daytime O3 in the spring and an overestimation of daytime concentrations during the summer[4, 5]. Evaluation of CMAQ to the Ozone Monitoring Instrument (OMI) found that CMAQ overestimates summertime nitrogen dioxide (NO2) in urban locations and underestimates in rural locations[6].”
Comment: The English writing should be improved as well.
Response: Multiple revisions to the English writing has been made throughout the manuscript.
References:
Zhang, Y., et al., A measurement‐model fusion approach for improved wet deposition maps and trends. Journal of Geophysical Research: Atmospheres, 2019. Agency, U.E.P., National Emissions Inventory (NEI), Facility-level, US, 2005, 2008, 2011. Appel, K.W., et al., Evaluation of the community multiscale air quality (CMAQ) model version 4.5: sensitivities impacting model performance; part II—particulate matter. Atmospheric Environment, 2008. 42(24): p. 6057-6066. Appel, K.W., et al., Examination of the Community Multiscale Air Quality (CMAQ) model performance over the North American and European domains. Atmospheric Environment, 2012. 53: p. 142-155. Appel, K.W., et al., Evaluation of the Community Multiscale Air Quality (CMAQ) model version 4.5: sensitivities impacting model performance: part I—ozone. Atmospheric Environment, 2007. 41(40): p. 9603-9615. Canty, T., et al., Ozone and NOx chemistry in the eastern US: evaluation of CMAQ/CB05 with satellite (OMI) data. Atmos. Chem. Phys, 2015. 15(19): p. 10965-10982.

Round 2
Reviewer 1 Report
See attached file.

Reviewer 2 Report
The authors did a good job of responding to most of my concerns/suggestions with the exception of Comment #1.
Comment #1 - Suggestion to include data from the IMPROVE network
I am troubled that the authors decided to ignore the availability of a large data set which could significantly reduce the data paucity problem (especially in the western US). I recognize that inclusion of this data set into the fusion method would be time consuming. However, I believe that it would likely have a significant impact on the quality of the fused data set. If the goal of the manuscript is to demonstrate the viability of the data fusion method then it may not be necessary to include the data from the IMPROVE network. If, on the other hand, the goal is to create a fused data set which can be used by other researchers for exposure assessment purposes (as implied in the abstract) then I think that inclusion of the additional data is crucial. The manuscript clearly shows that the CMAQ model has significant biases which must be corrected and the quality of the correction depends critically upon the density of monitoring stations in a given region.
I strongly suggest that the authors include the IMPROVE data in the fusion method and then make the fused data set publicly available through one of several data repositories for use by other researchers. Such a fused data set would be extremely valuable because the uncertainties will have already been quantified. I believe that such an article, with a well-characterized, attached data set, would be well utilized and cited. A methodology manuscript without a publicly-available fused data set would be only moderately interesting and would be cited much less frequently.
Comment #2 - Description of CMAQ biases
Satisfactory response
Comment #3 - No metadata for CMAQ simulations
Satisfactory response. However, there is one error on lines 89 and 90. The text states "35 vertical layers that span till the top of the free troposphere, each layer with a height of 19 m". It should read "35 vertical layers that span till the top of the free troposphere, with layer 1 nominally 19 m tall" (language taken from reference 37)
Comment #4 - No justification for power law regression
Satisfactory response
Comment #5 - East vs West performance
Satisfactory response.
Comment #6 - References needed for health impacts
Satisfactory response.
Minor Comments (New)
Line 278 - The statement that "O3, PM10, and SO2 were found to have the greatest increase in performance temporally" is not supported by Figure 3. I agree with PM10 and SO2, but not O3.
Lines 279-280 - "These species (O3, PM10 and SO2) had an increase in temporal R2 of 0.70" is not supported by Figure 3. I agree with PM10 and SO2, but not O3.
Reviewer 3 Report
The authors adequately responded to the reviewer’s comments on this study. The proposed data fusion method for combining observations and CTM simulations is more complete and clearer in the revision. The results of improvement are also acceptable. Thus, I have only one question which still unclear in the current manuscript. That is, most of the observations (stations) collected for the data fusion and validation are from the surface layer, while the simulations of CTM (CMAQ) are implemented in a 3 dimensional domain. Suppose the fusion procedure proposed in this study is sorely focused on the surface layer. Therefore, the layer of CMAQ simulation used for the fusion procedure should be identified in the Section of materials and methods. On the other hand, the pollutants in vertical domain may include the transported pollutants above the surface layer in addition to the surface emissions. It’s still essential to validate the spatial distribution for the performance of fusion results. Due to the limitation in the comparison between gridded simulation and point observation, the satellite retrievals of pollutants are suggested accordingly.
